# TRANSFORMER-QL: A STEP TOWARDS MAKING TRANSFORMER NETWORK QUADRATICALLY LARGE

## ABSTRACT

Transformer networks have shown outstanding performance on many natural language processing tasks. However the context length (the number of previous tokens on which the output states depend) of a Transformer network grows at best linearly with the memory and computational power used. This limitation prevents a transformer network to have very long context in a resource limited application. In this work, we propose a class of transformer networks, namely Transformer-QL (**Q**uadratically **L**arge), in which, the context length can grow at best quadratically with the memory and computational power used. We have empirically evaluated a Transformer-QL model in three long range language modeling datasets. The results show that Transformer-QL can provide significant improvements over other state of the art networks.

## 1 INTRODUCTION

Since its introduction in Vaswani et al. (2017), Transformer networks have overtaken its predecessor Recurrent Neural Networks (RNN) in almost every natural language processing task. However, one limitation of Transformer network is its high requirement of memory and computational power. In a vanilla Transformer network, the memory and computational requirement grows quadratically with the sequence length, and thus with context length.

In an effort to overcome the above limitation, Transformer-XL (Dai et al., 2019) and Compressive Transformer (Rae et al., 2020) have been recently proposed. However, in both the network, the context length can grow at best linearly with the memory and computation usage. An alternative strategy have been explored in Li et al. (2019); Ye et al. (2019); Child et al. (2019); Zaheer et al. (2020); Beltagy et al. (2020); Wang et al. (2020b); Kitaev et al. (2020); Katharopoulos et al. (2020); Choromanski et al. (2020); Wang et al. (2020a). All these works have proposed to replace the vanilla self-attention network by a different one with linear or log-linear memory and computation complexity leading to novel transformer architectures with overall linear or log-linear cost. Although, they provide an improvement over the quadratic cost of the vanilla transformer network, the achieved cost is still, at best, linear. Besides, since those techniques are based on either sparsification or compression of the self attention mechanism, they struggle to accumulate long distance information (Gupta & Berant, 2020).

Several works such as Burtsev & Sapunov (2020); Ainslie et al. (2020); Gupta & Berant (2020) have proposed to increase the context length by introducing a global attention which attains to every input token, thus capable of capturing long distance dependency. However, capturing long distance dependency using those approaches involves extreme compression of state space by the global attention mechanism. Moreover, even though, they perform well on several tasks, their performance on language modeling task have not been tested. Another line of work (Zhang et al., 2019; Pappagari et al., 2019) have suggested to use hierarchical arrangement of transformer network to capture document-wide dependency. However, applicability of those networks requires hierarchical structure in the data itself. Moreover, those techniques have been proposed for document compression rather than language modeling.

In this paper, we propose a class of transformer architectures, namely Transformer-QL (**Q**uadratically **L**arge), to alleviate the problem of capturing long distance dependency. Similar to multi-scale transformer networks (Donahue et al., 2019; Subramanian et al., 2020; Zhao et al., 2020;

Dai et al., 2020), Transformer-QL captures the contextual information in multiple temporal scales - finer scales to capture recent past information and coarser scales to capture distance past information. Additionally like Transformer-XL, it keeps the hidden states of a past segment in memory and use it to process future segments causing the context length to grow beyond the current segment. Overall, the context length in Transformer-QL can grow up to quadratically with the memory/computational usage. The contributions of the work are as follows:

- We have proposed a novel class of transformer architectures, namely Transformer-QL, in which, the context length can be made to grow linearly with memory and computation cost. Further, employing a linear cost self attention layer like Wang et al. (2020b); Katharopoulos et al. (2020), the context length of Transformer-QL can be made to grow quadratically in both memory and computational cost.

- We have empirically evaluated a Transformer-QL model on three long range language modeling datasets. The results show significant improvement in perplexity score over Transformer-XL and Compressive Transformer.

The organization of the paper is as follows. In section 2, the proposed Transformer-QL architecture along with its background has been introduced. Section 3 provides empirical evaluation of Transformer-QL. The section also studies the sensitivity of Transformer-QL to several hyperparameters. Finally, in Section 4, the conclusion has been drawn and future directions of the work have been suggested.

## 2 METHOD

### 2.1 TERMINOLOGY AND NOTATIONS

In a transformer network, the input sequence are partitioned into smaller segments of fixed length. Each segment is processed independently of other segments. We refer to the number of tokens in each segment as *segment length*. In a transformer network with recurrent memory like Transformer-XL, the hidden states of the recent past segments are preserved in a fixed length memory. We refer to the number of tokens in each layer of the memory unit as *memory length*. For an output state (i.e. the output states of the last layer), we use the term *context length* to refer to the number of past tokens on which the output state depends. In transformer network, different output states might have different context length. We respectively refer the minimum and maximum of the context lengths of all the output states in a network as *minimum context length* and *maximum context length* of the network. We refer the sum of segment length and the memory length using the term *window length*.

We denote the segment length, memory length, window length and model dimension by $n_s, n_m, n_w$ and $d_m$ respectively. Thus, we have $n_w = n_s + n_m$. We also use the notations $\mathbf{s}_t^l$ and $\mathbf{m}_t^l$ to denote the output and memory of $l$-th layer at time step $t$ for $l = 1, 2, \cdots, L$ where $L$ is the total number of Layers. The output and memory of embedding layer at time step $t$ have been denoted by $\mathbf{s}_t^0$ and $\mathbf{m}_t^0$ respectively. The number of heads in the self attention layers has been denoted by $H$.

### 2.2 BACKGROUND

**Transformer** A transformer network consists of stacked collection of multiple transformer layers. Each transformer layer contains a multi-head self-attention layer followed by a position-wise feed forward layer. Though the memory and computational cost of position-wise feed forward layer is linear in the length of input sequence, the multi-head self attention layer has a quadratic cost. The transformer network tackles the quadratic memory and computational cost by dividing the input sequence into smaller segments and applying the transformer network on each segment independently. However, such method limits the context lengths to the segment length. Dai et al. (2019) has named this problem as context fragmentation problem.

**Transformer-XL** Dai et al. (2019) has proposed Transformer-XL to solve the context fragmentation problem. In Transformer-XL, instead of discarding the hidden states after the computation of a segment, they are saved in memory (please refer to Figure 3). During the computation of the following segments, the self attention is applied over the hidden states of both the current segment and the

memory, thus has an increased context length without quadratic increase in the memory and computational cost. In fact, the memory/computational cost of the self-attention layer of Transformer-XL grows only quadratically only with the segment size $n_s$ and linearly with the memory length $n_m$. On the other hand, the context lengths get increased by a length of $n_m$ per layer. By keeping $n_s$ small and making $n_m$ large enough, the memory and computational cost of Transformer-XL can be made close to linear with respect to the context lengths. Rae et al. (2020) has proposed to improved the memory/computational cost of Transformer-XL further by keeping the part of the memory states in a compressed form. However, even with this improvement, the memory and computational cost can be at best linear in the context length.

## 2.3 THE MODEL

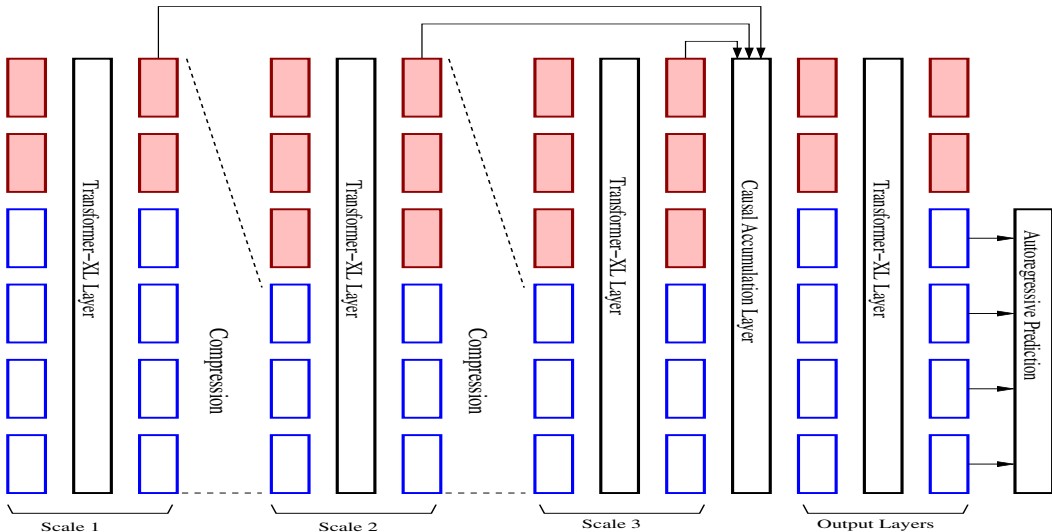

Figure 1: High Level Visualization of Proposed Model. The model processes the input tokens in multiple temporal scales. Each scale has several transformer layers with recurrent memory. The output of the last layer of one scale is compressed to form the input of the next scale. As the segment length gets reduced because of compression, the memory length is increased to make the total length (i.e. segment length + memory length) of all the layers same. In the figure, the blue boxes represent hidden states of the current time step where as the red boxes represent the memory states.

**Overview**  In this paper, we explore to increase the context length by compressing the hidden states hierarchically. The high level view of our architecture is shown in Figure 1. As shown in the figure, the model processes the input sequence in several scales of temporal granularity. Each scale consists of several transformer layers with recurrent memory. The output of the last layer of one scale is compressed causing the temporal granularity as well as the segment length to reduce. As the segment length reduces, we also simultaneously increase the memory length to keep the total length (i.e. segment length + memory length) of the layer constant. Then the new segment and memory is fed as the input to the first layer of the next scale. The resulting architecture is similar to the multi-scale transformer architectures (Donahue et al., 2019; Subramanian et al., 2020; Zhao et al., 2020). Additionally, Transformer-QL keeps recurrent memory to store hidden states of previous segments. Therefore, in Transformer-QL, the layers belonging to a finer scale process contextual information in fine-grained manner, but have a smaller context length. On the other hand, a layer belonging to a coarser scale process information in coarse-grained manner, but have a longer context length (please refer to Figure 5 for a detailed illustration of the context lengths of Transformer-QL layers). To get the final output of the network, we causally combine the (possibly over-sampled) outputs of the last layer of each scale and pass those through several transformer layers (following Subramanian et al. (2020); Zhao et al. (2020)) to learn deep representation of the output.

**Function** Compute(**s**, **m**, $l$)

1 **begin**
2     $\mathbf{s}' \leftarrow \text{MultiHeadSAttn}^l(\mathbf{s}, \mathbf{m})$
3     $\mathbf{s}'' \leftarrow \text{LayerNorm}(\mathbf{s} + \mathbf{s}')$
4     $\mathbf{s}''' \leftarrow \text{LayerNorm}(\mathbf{s}'' + \text{PoswiseFF}^l(\mathbf{s}''))$
5     **return** $\mathbf{s}'''$

(b) One Transformer-XL layer over hidden states **s** and memory **m**

**Function** Shift(**s**, **m**)

1 **begin**
2     $n_m \leftarrow \text{len}(\mathbf{m})$
3     $\mathbf{m}' \leftarrow \text{concat}(\mathbf{m}, \mathbf{s})$
4     $\mathbf{m}'' \leftarrow \mathbf{m}'[-n_m :]$
5     **return** $\text{stop\_gradient}(\mathbf{m}'')$

(d) Shift the current hidden state **s** into memory **m**

1 At the beginning
2 **begin**
3     $\mathbf{m}_0^0, \cdots, \mathbf{m}_0^{L-1} \leftarrow \mathbf{0}$         // initialize all the memory to zero
4     $t \leftarrow 1$               // initialize time step to 1
5 **while** *there is more data to process* **do**
6     $\mathbf{s}_t^0 \leftarrow \mathbf{x}_t \mathbf{W}_{emb}$            // embed input segment
7     $\mathbf{m}_t^0 \leftarrow \text{Shift}(\mathbf{s}_t^0, \mathbf{m}_{t-1}^0)$       // shift hidden states into memory
8     shift_len $\leftarrow n_s^0$          // initialize shift length to $n_s^0$
9     acc_inputs $= [\,]$     // list to hold all the inputs of accumulation layer
10    $l_g \leftarrow 1$             // initialize global layer index to 1
11    **for** $i \leftarrow 1$   to   *no_of_scales* **do**
12       $L_i \leftarrow$ no of layers at scale $i$    // set $L_i$ to the number of layers at scale $i$
13       **for** $l \leftarrow 1$   to   $L_i$ **do**
14         $\mathbf{s}_t^{l_g} \leftarrow \text{Compute}(\mathbf{s}_t^{l_g-1}, \mathbf{m}_{t-1}^{l_g-1}, l_g)$     // run one Transformer-XL layer
15         $\mathbf{m}_t^{l_g} \leftarrow \text{Shift}(\mathbf{s}_t^{l_g}[: \text{shift\_len}], \mathbf{m}_{t-1}^{l_g})$    // shift hidden states into memory
16         $l_g \leftarrow l_g + 1$        // increase the global layer index
17       acc_inputs.append($[\mathbf{s}_t^{l_g-1}, \mathbf{m}_t^{l_g-1}]$)     // put output of scale $i$ in acc_inputs
18       **if** $i <$ *no_of_scales* **then**
         // not the last scale, add a compression layer
19         $\mathbf{s}_t^{l_g} \leftarrow f_c(\text{concat}(\mathbf{m}_{t-1}^{l_g-1}, \mathbf{s}_t^{l_g-1}))[-n_s^{l_g} :]$       // compress the states
20         shift_len $\leftarrow$ shift_len$/c$     // reduce the shift_len for the new scale
21         $\mathbf{m}_t^{l_g} \leftarrow \text{Shift}(\mathbf{s}_t^{l_g}[: \text{shift\_len}], \mathbf{m}_{t-1}^{l_g})$    // shift hidden states into memory
22         $l_g \leftarrow l_g + 1$        // increase the global layer index
23    shift_len $\leftarrow n_s^0$        // set the shift length for the output layers
24    $\mathbf{s}_t^{l_g} \leftarrow \text{Accumulate}(\text{acc\_inputs})$       // combine the states in acc_inputs
25    $\mathbf{m}_t^{l_g} \leftarrow \text{Shift}(\mathbf{s}_t^{l_g}[: \text{shift\_len}], \mathbf{m}_{t-1}^{l_g})$     // shift hidden states into memory
26    $l_g \leftarrow l_g + 1$          // increase the global layer index
27    $L_o \leftarrow$ no of output layers      // set $L_o$ to the number of output layers
28    **for** $l \leftarrow 1$   to   $L_o$ **do**
29       $\mathbf{s}_t^{l_g} \leftarrow \text{Compute}(\mathbf{s}_t^{l_g-1}, \mathbf{m}_{t-1}^{l_g-1}, l_g)$       // one Transformer-XL layer
30       $\mathbf{m}_t^{l_g} \leftarrow \text{Shift}(\mathbf{s}_t^{l_g}[: \text{shift\_len}], \mathbf{m}_{t-1}^{l_g})$    // shift hidden states into memory
31       $l_g \leftarrow l_g + 1$        // increase the global layer index
32    $t \leftarrow t + 1$            // increase the time step

(e) Forward pass of Transformer-QL. $\mathbf{s}_t^l$ and $\mathbf{m}_t^l$ represents the hidden states and memory states of layer $l$ at time step $t$. $n_s^l$ and $n_m^l$ represent the segment length and memory length of $l$-th layer.

Figure 2: Transformer-QL Algorithm.

**The Compression Function** For compression, we use one of average pooling and max pooling with pool size and stride both equal to $c$ where $c$ is the rate by which we compress the states while transitioning from one scale to the next. Let $\mathbf{s}_t^l$ and $\mathbf{m}_t^l$ be the output and memory states of $l$-th layer with length $n_s^l$ and $n_m^l$ respectively. We apply the compression function on the concatenation of $\mathbf{m}_t^l$ and $\mathbf{s}_t^l$ to get the output $\mathbf{s}_t^{l+1}$ of length $n_s^{l+1} = (n_s^l + n_m^l)/c$ (for simplicity assume that $n_s^l + n_m^l$ is divisible by $c$). If $n_s^{l+1} > n_s^l$, we take the last $n_s^l$ elements of $\mathbf{s}_t^{l+1}$ to form the output of the

compression layer. Finally, we keep a recurrent memory $\mathbf{m}_t^{l+1}$ of length $n_s^l + n_m^l - n_s^{l+1}$ making $n_s^l + n_m^l = n_s^{l+1} + n_m^{l+1}$ to hold.

**The Memory Updates**    In Transformer-XL with segment length $n_s$ and memory length $n_m$, the segment of length $n_s$ is get shifted into the memory. In other words, the memory for the next time step is computed as $\mathbf{m}_{t+1}^l = \text{concat}(\mathbf{m}_t^l, \mathbf{s}_t^l)[-n_m :]$ for all layer $l$. However, in Transformer-QL, the granularity of layers belonging to different scales are different. More precisely, a segment of length $n_s^0$ belonging to scale 1 is compressed into a segment of length $n_s^0/c^{i-1}$ at a layer belonging to scale $i$. Thus, in Transformer-QL, we update the memory of a layer $l$ belonging to scale $i$ as $\mathbf{m}_{t+1}^l = \text{concat}(\mathbf{m}_t^l, \mathbf{s}_t^l[: n_h^i])[-n_m^i :]$ where $n_h^i = n_s^0/c^{i-1}$ and $n_m^i$ are the shift length and the memory length at scale $i$ for $i = 0, 1, \cdots$ respectively. The complete algorithm of Transformer-QL is shown in Figure 2.

**Droppath Regularization**    Since the output of the last layer of every scale is summed in the accumulation layer, the path through a higher scale forms to a deeper network while the path through a lower scale forms to a shallower network. Consequently, layers in the higher scales might remain under-fitted due to lack of gradient flow through the deeper network while the layers in the lower scales get over-fitted. To alleviate this problem, we have introduced *droppath* regularization. In the accumulation layer, let each output be computed as $s^o = \frac{1}{l}\sum_{i=1}^l s^i$ where $s^i$ represents the (possibly over-sampled) output of scale $i$ and $l$ is the total number of scales. In droppath regularization with *droppath probability* $p$, we drop the output of all the scales below $j$ with a probability $p/(l-1)$ for $j = 2, 3, \cdots, l$ from the accumulated output. More precisely, we generate a random number $u$ from uniform probability distribution and compute the output as $s^o = \frac{1}{l-j+1}\sum_{i=j}^l s^i$ if $u \in \left[\frac{(j-2)p}{l-1}, \frac{(j-1)p}{l-1}\right]$. For $u \geq p$, no droppath is applied.

## 2.4    THE COMPLEXITY

The memory/computational complexity of a Transformer-XL network (Dai et al., 2019) with segment length $n_s$, memory length $n_m$ and $L$ layer is $\Theta((\alpha(n_m, n_s) + n_s)L)$ where $\alpha(\cdot, \cdot)$ is the complexity of self-attention layer. The context length $n_c$ of the network is $\Theta(n_m L)$. Since $\alpha(n_m, n_s) = \Omega(n_m + n_s)$ (Li et al., 2019; Ye et al., 2019; Child et al., 2019; Zaheer et al., 2020; Beltagy et al., 2020; Wang et al., 2020b; Kitaev et al., 2020; Katharopoulos et al., 2020; Choromanski et al., 2020), the memory and computational complexity of Transformer-XL in terms of context length is $\Omega(n_c)$. Similarly, the memory and computational complexity of Compressive Transformer (Rae et al., 2020) in term of context length is $\Omega(n_c/c)$ where $c$ is the compression rate. Therefore, the memory/computational complexity of both Transformer-XL network and Compressive Transformer network in term of the context length is at least linear. Consequently, increasing the context length in both the networks requires at least linear increase in the amount of both memory and computational requirements.

On the other hand, a Transformer-QL network with $L$ Transformer-XL layers and $i$ compression layers, the context length $n_c$ becomes $\Theta(c^i(n_s + n_m)) = \mathcal{O}(c^{\log_c n_s}(n_s + n_m)) = \mathcal{O}(n_s(n_s + n_m))$ where $n_s = n_s^0$ and $n_m = n_m^0$ are the segment and memory length in scale 1 of the network. Note that, since at most $i = \log_c n_s$ compression layer can be used in Transformer-QL, we have $c^i = \mathcal{O}(c^{\log_c n_s}) = \mathcal{O}(n_s)$. If we set $n_m = \mathcal{O}(n_s)$, we have $n_c = \mathcal{O}((n_s)^2)$. However, the time and memory complexity of a Transformer-QL network is $\Theta(\alpha(n_s, n_m)L + (n_s + n_m)i)) = \Theta(\alpha(n_s, n_m)(L + i))$. Since $\alpha(n_s, n_m) = \Omega(n_s + n_m)$ and we set $n_m = \mathcal{O}(n_s)$, the memory/computational complexity of Transformer-QL becomes $\Omega(n_s(L + i))$. Therefore, the memory/computational complexity of Transformer-QL in terms of context length is $\Omega(\sqrt{n_c}(L + i)) = \Omega\left(\sqrt{n_c}(L + \log_c n_s)\right)$. Thus, the complexity of Transformer-QL can be at best sub-linear. Moreover, if we set the compression rate $c$ to $n_s$, the memory and computational complexity can be at best $\Theta(\sqrt{n_c})$ or, in other words, the context length can be at best quadratic in the memory and computational cost. In Appendix B, we provide an algorithm to compute a tight estimation of the context length of a Transformer-QL network. In the appendix, we have also provided a detailed illustration of the dependency structure of the hidden states of a Transformer-QL network on the past tokens.

| Dataset | Model | Test | | Average | Test |
| --- | --- | --- | --- | --- | --- |
| | | $n_s/n_m/n_{cm}$ | $n_w$ | test $n_c$ | Perplexity |
| SimpleBooks-2 | Transformer-XL | $04/12/-$ | 16 | 98 | 20.15 |
| | Comp-Transformer | $04/06/06$ | 16 | 146 | 19.67 |
| | Transformer-QL | $04/12/-$ | 16 | 138 | **18.78** |
| | Transformer-XL | $08/24/-$ | 32 | 196 | 19.61 |
| | Comp-Transformer | $08/12/12$ | 32 | 292 | 19.15 |
| | Transformer-QL | $08/24/-$ | 32 | 276 | **18.56** |
| SimpleBooks-92 | Transformer-XL | $04/12/-$ | 16 | 98 | 12.93 |
| | Comp-Transformer | $04/06/06$ | 16 | 146 | 12.49 |
| | Transformer-QL | $04/12/-$ | 16 | 138 | **12.17** |
| | Transformer-XL | $08/24/-$ | 32 | 196 | 12.35 |
| | Comp-Transformer | $08/12/12$ | 32 | 292 | 12.01 |
| | Transformer-QL | $08/24/-$ | 32 | 276 | **11.88** |
| WikiText-103 | Transformer-XL | $04/12/-$ | 16 | 98 | 31.19 |
| | Comp-Transformer | $04/06/06$ | 16 | 146 | 31.91 |
| | Transformer-QL | $04/12/-$ | 16 | 138 | **29.13** |
| | Transformer-XL | $08/24/-$ | 32 | 196 | 27.52 |
| | Comp-Transformer | $08/12/12$ | 32 | 292 | 27.19 |
| | Transformer-QL | $08/24/-$ | 32 | 276 | **26.63** |

Table 1: Perplexity scores (lower is better) of the three networks: Transformer-QL, Transformer-XL and Compressive Transformer (Comp-Transformer). The third column shows the segment length ($n_s$), memory length ($n_m$) and compressed memory length ($n_{cm}$) of the test model. The forth column shows the window length $n_w$ of the test model. Note that, for Transformer-XL and Transformer-QL, the window length is $n_s + n_m$ and for Compressive Transformer the window length is $n_s + n_m + n_{cm}$. The average test context length $n_c$ has been shown in the fifth column. The average test context length has been computed by taking the average of the minimum and maximum context length of the test model where the minimum and maximum context length have been calculated using the algorithm of Appendix B.

## 3 EMPIRICAL EVALUATION

In this section, we empirically evaluate the efficacy of Transformer-QL for long range language modeling task. Towards that goal, we compare the results of Transformer-QL with that of Transformer-XL (Dai et al., 2019) and Compressive Transformer (Rae et al., 2020). Then we evaluate the sensitivity of Transformer-QL to several hyper-parameters.

### 3.1 COMPARISON WITH STATE OF THE ART METHODS

**State of the Art Methods** We compare Transformer-QL network with the following two networks:

**Transformer-XL (Dai et al., 2019)** Transformer-XL is similar to vanilla Transformer with two modifications. It uses recurrent memory to store and access the hidden states of the past time steps. The recurrent memory enables to increase the minimum context length up to $n_m L$ where $n_m$ is the memory length and $L$ is the number of layers. It also uses relative positional embedding of token instead of absolute positional embedding.

**Compressive Transformer (Rae et al., 2020)** Like Transformer-XL, Compressive Transformer also uses recurrent memory. However, Compressive Transformer keeps part of the recurrent memory in compressed format, thus has an increased context length over Transformer-XL.

**Datasets** We compare Transformer-QL against the above two networks on three long range language modeling datasets: SimpleBooks-2 (Nguyen, 2019), SimpleBooks-92 (Nguyen, 2019) and WikiText-103 (Merity et al., 2017). SimpleBooks-2 and SimpleBooks-92 are created from Gutenberg book corpus (www.gutenberg.org) while WikiText-103 are created from Wikipedia articles. All

| Model dimension | Train $n_s/n_m$ | Test $n_s/n_m$ | Trans-QL | Trans-XL | Improvement | Relative Improvement |
|---|---|---|---|---|---|---|
| 512 | 16/16 | 16/16 | 33.32 | 32.68 | −0.64 | −1.96% |
| 1024 | 16/16 | 16/16 | 27.70 | 28.42 | 0.72 | +2.53% |
| 1536 | 16/16 | 16/16 | 27.00 | 28.22 | 1.22 | +**4.32**% |

Table 2: Improvement in test perplexity score (lower is better) of Transformer-QL over Transformer-XL for three different model dimensions. The forth and fifth columns show the test perplexity obtained by Transformer-QL and Transformer-XL respectively.

the three datasets preserve paragraph and section structures of their sources making those suitable for long range language modeling task. The statistics of the three datasets are shown in Table 4 of Appendix C.

**Experimental Details**   For the experiments of Transformer-XL and Compressive Transformer, we have used an 8-layer network. And for the experiments of Transformer-QL, we have used a network with 3 layers in scale 1, 3 layers in scale 2 and 2 layers in the output block. Thus, the Transformer-QL has a total of eight layers as in Transformer-XL and Compressive Transformer. We set the compression rate of Compressive Transformer to 2. In Transformer-QL, we have used max-pooling layer with pool size 2 and stride 2 as the compression layer. Thus, both the Transformer-QL and Compressive Transformer have a compression rate of 2. For the experiments on the SimpleBooks-92 and WikiText-103, we have set the model dimension to 1536 and used an initial learning rate of $1 \times 10^{-4}$. On the other hand, for the experiments on SimpleBooks-2, we have set the model dimension to 256 and learning rate to $2.5 \times 10^{-4}$. All the models have been trained using Adam optimizer. We set the droppath probability of Transformer-QL to 0.3. The details of other hyper-parameters can be found in Appendix E .

**Results**   The results of the comparison are shown in Table 1. The results are grouped by the window length $n_w$ of the test model as the lower bound of memory and computation requirement directly depends on it. In all the datasets and settings, Transformer-XL performs worst among all three. Worst performance of Transformer-XL is not surprising as it has smallest average context length (shown in the fifth column) for a given $n_w$. However, Compressive Transformer has a slightly larger average context length than Transformer-QL. Yet, Transformer-QL has performed similarly or significantly better than Compressive Transformer in all the setting which indicates that Transformer-QL can exploits the contextual information more effectively than Compressive Transformer.

### 3.2   EFFECT OF MODEL DIMENSION

In this section, we investigate the effect of model dimension on the performance of Transformer-QL. Towards that goal, we have performed experiments on WikiText-103 dataset with varying model dimension. For each experiment, we compared the test perplexity of Transformer-QL with that of Transformer-XL. The results are shown in Table 2. The *improvement* in test perplexity of Transformer-QL over Transformer-XL has been computed by subtracting the test perplexity of Transformer-QL from that of Transformer-XL. The *relative improvement* is computed by

$$Relative\ improvement = \frac{Improvement \times 100}{Test\ perplexity\ of\ Transformer\text{-}XL}$$

As shown in the table, Transformer-QL performs relatively worse for small model dimension like 512 and relative improvement increases as the model dimension increases. We speculate that the relatively worse performance of Transformer-QL for smaller model dimension is caused by the difficulty in compressing hidden states during switching from one scale to the next. To alleviate the problem, Donahue et al. (2019) have proposed to increase the model dimension as the model transits from a lower scale to a higher one. On the other hand, Dai et al. (2020) have suggested a novel query-only-pooling to solve the problem. We take it as a future work to try those approaches in Transformer-QL.

## 3.3 Effect of Context Length

In this section, we study the relative improvement in perplexity scores obtained by Transformer-QL over Transformer-XL for varying context length. The results are shown in Table 3. From the table, it can be noticed that relative improvement obtained by Transformer-QL is more when the context length of the Transformer-XL networks are smaller in the first place. For example, for test $n_s = 08$ and $n_m = 08$, the relative improvement is as high as $8.80\%$. On the other hand, for the test $n_s = 02$ and $n_m = 30$, the relative improvement is only $2.76\%$. This can be explained by the fact that for the segment and memory length $02$ and $30$, the average context length of Transformer-XL is already large enough (241) to provide good result. By extending the average context length from 241 to 332, Transformer-QL provides only a small improvement following the law of diminishing return (Hestness et al., 2017).

| Model Dimension | Test $n_s/n_m$ | Avg. test $n_c$ | | Perplexity | | |
|---|---|---|---|---|---|---|
| | | Trans-XL | Trans-QL | Trans-XL | Trans-QL | Rel Imprv |
| 1536 | 08/08 | 68 | 100 | 33.08 | 30.17 | +8.80% |
| | 16/16 | 136 | 200 | 28.22 | 27.00 | +4.32% |
| | 02/30 | 241 | 332 | 27.19 | 26.44 | +2.76% |

Table 3: Relative improvements in perplexity scores (lower is better) obtained by Transformer-QL over Transformer-XL on WikiText-103 dataset. The second column shows the test segment ($n_s$) and memory ($n_m$) length. The third and forth column respectively show the average test context length ($n_c$) of Transformer-XL and Transformer-QL network.
.

## 4 Conclusion and Future Work

In the work, we have proposed a class of transformer networks namely Transformer-QL in which the context length can grow quadratically in memory and computational usage. Our empirical evaluation shows that Transformer-QL can perform significantly better than other long range language modeling networks like Transformer-XL and Multi-scale Transformer by exploiting longer context length. Further more, it can perform significantly better than Compressive Transformer by exploiting the contextual information more effectively.

In our empirical evaluation, we have evaluated a Transformer-QL network with only one compression layer. In future, we want to evaluate a network with more then one compression layers. Also, we have empirically found that the performance of Transformer-QL network can be worse than that of Transformer-XL when the model dimension is small. As our future work, we want explore different methods for removing this limitation.

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

## A  TRANSFORMER-XL ALGORITHM

The algorithm for Transformer-XL is shown in Figure 3.

```
1  At the beginning
2  begin
3  │   m_0^0, ⋯, m_0^{L-1} ← 0              // initialize memory to zero
4  │   t ← 1                                // initialize time step to 1
5  while there is more data to process do
6  │   s_t^0 ← x_t W_{emb}                  // embed input segment
7  │   m_t^0 ← Shift(s_t^0, m_{t-1}^0)      // shift hidden states into memory
8  │   for l ← 1 to L do
9  │   │   s_t^l ← Compute(s_t^{l-1}, m_{t-1}^{l-1}, l)   // run one Transformer-XL layer
10 │   │   m_t^l ← Shift(s_t^l, m_{t-1}^l)  // shift hidden states into memory
11 │   t ← t + 1                            // increase the time step
```

Figure 3: Forward pass of Transformer-XL. The functions *Compute* and *Shift* are given in Figure 2b and 2d respectively.

```
1   cur_slen ← n_s                          // set current segment length to n_s
2   cur_mlen ← n_m                          // set current memory length to n_m
3   min_clen ← 0                            // initialize minimum context length to 0
4   cur_layer_clen ← n_m                    // set current layer context length to n_m
5   for i ← 1 to no_of_scales do
6   │   L_i ← no of layers at scale i       // set L_i to the number of layers at scale i
7   │   for l ← 1 to L_i do
8   │   │   min_clen ← min_clen + cur_layer_clen   // increase min_clen by cur_layer_clen
9   │   if i < no_of_scales then
        │   // there is a compression layer
        │   // update min_clen, cur_mlen, cur_slen and cur_layer_clen
        │   // appropriately
10  │   │   min_clen ← min_clen + min(cur_mlen, n_s)/cur_slen
11  │   │   cur_mlen ← (n_m + n_s) - min((n_m + n_s)/2, n_s)
12  │   │   cur_slen ← cur_slen/c
13  │   │   cur_layer_clen ← cur_mlen × n_s/cur_slen
14  cur_layer_clen ← n_m                    // reset cur_layer_clen for output layers
15  L_o ← no of output layers               // set L_o to the number of output layers
16  for l ← 1 to L_o do
17  │   min_clen ← min_clen + cur_layer_clen   // increase min_clen by cur_layer_clen
18  return min_clen
```

Figure 4: Computation of minimum context length of a Transformer-QL model. The $n_s$, $n_m$ respectively represent the segment and memory length of the scale 1 of the network.

## B  CONTEXT LENGTH OF TRANSFORMER-QL

A tight estimate of the minimum context length of a Transformer-QL network can be computed using algorithm of Figure 4. For simplicity, we have assumed that all the division operations result into integer output. We have also assumed that there is at least one layer in every scale. The maximum context length can be obtained by adding $n_s$ to the minimum context length.

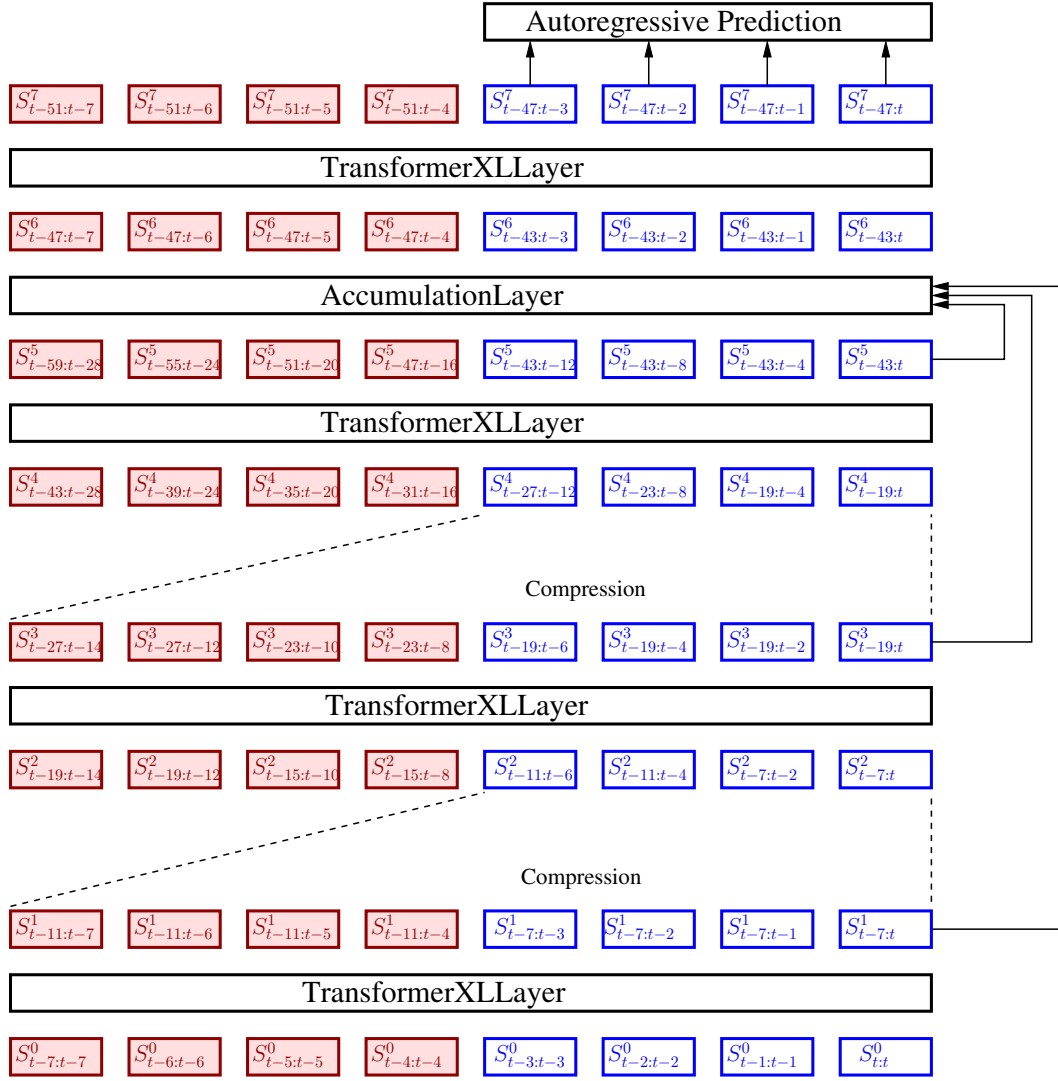

Figure 5: Dependency of hidden states to the past tokens in a Transformer-QL network.

Additionally, in Figure 5, we have shown the detailed computation of minimum/maximum context length with an example. In the figure, the notation $S^l_{t_1:t_2}$ used to denote a hidden states of $l$-th layer and the state depends on the $t_1$-th to $t_2$-th tokens of the input sequence. In the example of the figure, each output state depends on at least $44$ previous tokens. In other words, minimum context length of the network is $44$. On the other hand, in a Transformer-XL network of same segment length, memory length and number of layers, the minimum context length would have been $4 \times 4 = 16$.

## C  STATISTICS OF DATASETS

The statistics of the datasets are shown in Table 4.

| Dataset | Number of Tokens | Vocabulary Size | Average Frequency |
|---|---|---|---|
| SimpleBooks-2 | 2.2M | $11,492$ | 195.43 |
| SimpleBooks-92 | 91.5M | $98,304$ | 931.40 |
| WikiText-103 | 103M | $267,735$ | 385.56 |

Table 4: Statistics of the datasets used in the experiments.

## D  COMPARISON WITH MULTI-SCALE TRANSFORMER

In this section, we empirically compare Transformer-QL with Multi-scale Transformer (Subramanian et al., 2020). Our implementation of Multi-scale Transformer is same as Transformer-QL without any recurrent memory. The resultant Multi-scale Transformer is similar to the button-up model of Subramanian et al. (2020). We have used hyperparameter settings same as Transformer-QL to train the Multi-scale Transformer. The result is shown in Table 5. From the table, we can see that Multi-scale Transformer has been widely bitten by Transformer-QL even when the Multi-scale Transformer has been trained and tested with a larger window length.

| Dataset | Model | Train | | Test | | | Average | Test |
|---|---|---|---|---|---|---|---|---|
| | | $n_s$ | $n_m$ | $n_s$ | $n_m$ | $n_w$ | test $n_c$ | Perplexity |
| SimpleBooks-2 | MS-Transformer | 16 | — | 16 | — | 16 | 8 | 25.13 |
| | Transformer-QL | 08 | 08 | 08 | 08 | 16 | 100 | **18.92** |
| | MS-Transformer | 16 | — | 64 | — | 64 | 32 | 21.89 |
| | Transformer-QL | 08 | 08 | 16 | 16 | 32 | 200 | **18.57** |
| SimpleBooks-92 | MS-Transformer | 64 | — | 16 | — | 16 | 8 | 28.37 |
| | Transformer-QL | 08 | 08 | 08 | 08 | 16 | 100 | **12.42** |
| | MS-Transformer | 64 | — | 64 | — | 64 | 32 | 14.30 |
| | Transformer-QL | 08 | 08 | 16 | 16 | 32 | 200 | **11.92** |
| WikiText-103 | MS-Transformer | 128 | — | 32 | — | 32 | 16 | 44.05 |
| | Transformer-QL | 16 | 16 | 08 | 08 | 16 | 100 | **30.17** |
| | MS-Transformer | 128 | — | 128 | — | 128 | 64 | 29.15 |
| | Transformer-QL | 16 | 16 | 16 | 16 | 32 | 200 | **27.00** |

Table 5: Comparison of Transformer-QL with Multi-scale Transformer (MS-Transformer). The third and forth column respectively show the segment length ($n_s$) and the memory length ($n_m$) used during training. The fifth, sixth and seventh columns respectively show the segment, memory and the window ($n_w = n_s + n_m$) length used to compute the text perplexities. The eighth column shows the average context length ($n_c$) of the test models.

## E  HYPERPARAMETER SETTING

We used the following values for hyperparameter for the experiments on SimpleBooks-2 datasets:

| Hyperparameter | Transformer-XL | Compressive Transformer | Transformer-QL | Multi-scale Transformer |
|---|---|---|---|---|
| d_model | 256 | 256 | 256 | 256 |
| d_embed | 256 | 256 | 256 | 256 |
| div_val | 1 | 1 | 1 | 1 |
| untie_r | False | False | False | False |
| proj_same_dim | True | True | True | True |
| n_head | 4 | 4 | 4 | 4 |
| d_head | 64 | 64 | 64 | 64 |
| d_inner | 1024 | 1024 | 1024 | 1024 |
| train_batch_size | $\frac{128\times8}{n_s}$ | $\frac{128\times8}{n_s}$ | $\frac{128\times8}{n_s}$ | $\frac{128\times8}{n_s}$ |
| train $n_s$ | 08 | 04 | 08 | 08 |
| train $n_m$ | 08 | 06 | 08 | - |
| train $n_{cm}$ | - | 06 | - | - |
| pre_lnorm | True | True | True | True |
| warmup_steps | 0 | 0 | 0 | 0 |
| train_steps | 60,000 | 60,000 | 60,000 | 60,000 |
| learning_rate | 0.00025 | 0.00025 | 0.00025 | 0.00025 |
| min_lr_ratio | 0.004 | 0.004 | 0.004 | 0.004 |
| clip | 0.25 | 0.25 | 0.25 | 0.25 |
| dropout | 0.1 | 0.1 | 0.1 | 0.1 |
| dropatt | 0.1 | 0.1 | 0.1 | 0.1 |
| droppath | - | - | 0.3 | 0.3 |
| init_std | 0.02 | 0.02 | 0.02 | 0.02 |
| proj_init_std | 0.01 | 0.01 | 0.01 | 0.01 |
| recons_loss_weight | - | 0.01 | - | - |

For the SimpleBooks-92 datasets and model dimension 1536, the following values of hyperparameters are used:

| Hyperparameter | Transformer-XL | Compressive Transformer | Transformer-QL | Multi-scale Transformer |
|---|---|---|---|---|
| d_model | 1536 | 1536 | 1536 | 1536 |
| d_embed | 1536 | 1536 | 1536 | 1536 |
| div_val | 4 | 4 | 4 | 4 |
| untie_r | False | False | False | False |
| proj_same_dim | True | True | True | True |
| n_head | 16 | 16 | 16 | 16 |
| d_head | 96 | 96 | 96 | 96 |
| d_inner | 6144 | 6144 | 6144 | 6144 |
| train_batch_size | $\frac{512\times8}{n_s}$ | $\frac{512\times8}{n_s}$ | $\frac{512\times8}{n_s}$ | $\frac{512\times8}{n_s}$ |
| train $n_s$ | 08 | 04 | 08 | 08 |
| train $n_m$ | 08 | 06 | 08 | 08 |
| train $n_{cm}$ | - | 06 | - | - |
| pre_lnorm | True | True | True | True |
| warmup_steps | 0 | 0 | 0 | 0 |
| train_steps | 250,000 | 250,000 | 250,000 | 250,000 |
| learning_rate | 0.0001 | 0.0001 | 0.0001 | 0.0001 |
| min_lr_ratio | 0.004 | 0.004 | 0.004 | 0.004 |
| clip | 0.1 | 0.1 | 0.1 | 0.1 |
| dropout | 0.15 | 0.15 | 0.15 | 0.15 |
| dropatt | 0.15 | 0.15 | 0.15 | 0.15 |
| droppath | - | 0.3 | 0.3 | 0.3 |
| init_std | 0.02 | 0.02 | 0.02 | 0.02 |
| proj_init_std | 0.01 | 0.01 | 0.01 | 0.01 |
| recons_loss_weight | - | 0.01 | - | - |

For the WikiText-103 datasets and model dimension 1536, the following values of hyperparameters are used:

| Hyperparameter | Transformer-XL | Compressive Transformer | Transformer-QL | Multi-scale Transformer |
|---|---|---|---|---|
| d_model | 1536 | 1536 | 1536 | 1536 |
| d_embed | 1536 | 1536 | 1536 | 1536 |
| div_val | 4 | 4 | 4 | 4 |
| untie_r | False | False | False | False |
| proj_same_dim | True | True | True | True |
| n_head | 16 | 16 | 16 | 16 |
| d_head | 96 | 96 | 96 | 96 |
| d_inner | 6144 | 6144 | 6144 | 6144 |
| train_batch_size | $\frac{512\times16}{n_s}$ | $\frac{512\times16}{n_s}$ | $\frac{512\times16}{n_s}$ | $\frac{512\times16}{n_s}$ |
| train $n_s$ | 16 | 08 | 16 | 16 |
| train $n_m$ | 16 | 12 | 16 | - |
| train $n_{cm}$ | - | 12 | - | - |
| pre_lnorm | True | True | True | True |
| warmup_steps | 0 | 0 | 0 | 0 |
| train_steps | 350,000 | 350,000 | 350,000 | 350,000 |
| learning_rate | 0.0001 | 0.0001 | 0.0001 | 0.0001 |
| min_lr_ratio | 0.004 | 0.004 | 0.004 | 0.004 |
| clip | 0.1 | 0.1 | 0.1 | 0.1 |
| dropout | 0.15 | 0.15 | 0.15 | 0.15 |
| dropatt | 0.15 | 0.15 | 0.15 | 0.15 |
| droppath | - | - | 0.3 | 0.3 |
| init_std | 0.02 | 0.02 | 0.02 | 0.02 |
| proj_init_std | 0.01 | 0.01 | 0.01 | 0.01 |
| compression rate | - | 2 | 2 | 2 |
| recons_loss_weight | - | 0.01 | - | - |

For training models of model dimensions 512 and 1024, we have used initial learning rate of 0.0005 and 0.00025 respectively keeping the rest of the hyper-parameters same.

