# OpenReview forum: "Transformer-QL: A Step Towards Making Transformer Network Quadratically Large"
_ICLR.cc/2021/Conference — Reject_

### Official Review · AnonReviewer3 · 2020-10-28
**Initial review**

**Rating:** 5
**Confidence:** 3

**Review:**

Summary
-------
Using multi-scale hierarchical and compressive techniques, this paper examines a way to increase the context length of transformers.  Although the individual components are similar to previous work, they are combined in a novel way that shows a path toward longer and more efficient context lengths.

Positives
---------
Increasing the context length of transformers is an interesting and relevant topic, and the proposed solution can have real impact in moving the state of the art forward.

The results demonstrated in the experiments show an improvement over previous models.

The paper is clear and detailed, and well situated in the literature.  The algorithms were clear and the comments were useful for understanding the proposed idea.

Negatives
---------
The experiments do not compare to many other approaches, even though those approaches are cited throughout the paper.  In the conclusion, the paper mentions comparison with Multi-Scale approaches, but that is not present in the experiments.  Since the architecture (ignoring the compression) is similar to multi-scale approaches, it would be good to compare against empirically.

Although the complexity analysis is thorough, I'd like to see empirical results of memory/compute requirements as a function of the context length.  If I understand correctly, the sub-linear results depend on particular settings of the memory length and compression rate.  How do the settings used in the experiments compare to those used for the analysis?  Is there a perplexity/efficiency tradeoff, and can you characterize that experimentally?

Reasons for Score
-----------------
The idea proposed in the paper is novel and exciting, but I have some concerns about whether the gains promised by the theoretical analysis can be realized while maintaining modeling quality.

Minor issues that did not affect score
------------------
Figure 1 has some scaling/resolution issues that make it hard to read.  It's also sideways, which is inconvenient.

---

### Official Review · AnonReviewer2 · 2020-10-28

**Rating:** 5
**Confidence:** 4

**Review:**

The work introduces the Transformer-QL, a transformer-based model that aims to capture long distance dependencies in the input. The network processes the information defining multiple temporal scales, with finer scales for nearby elements, and coarser scales for distant information. It also includes the recurrent memory extension from Transformer-XL from Dai et al. The model is tested in a long range language modeling task.

==================

The presented Transformer-QL model improves over similar previous methods on language modeling. A thorough complexity analysis is included. The presentation can be improved, all the definitions are hard to follow. The layers are described in a textual fashion, barely any math (and extended in the pseudo-code). Further, the performance improvements are nice, though not impressive.

==================

The paper is hard to follow for moments. For example, a Figure to define what are $n_s, n_c, n_m$ would help to understand the paper with a nice visual benefit. Introducing the layer and networks in a simple way would help clarify the implementation and other notation.

The paper describes the idea of multiple temporal scales. In the experimental sections, it seems that only two scales are used. What would happen if the number of scales is increased to capture  longer contexts? Would the method gain in performance? Further, how does the network perform when a longer context is obtained *maintaining the same number of parameters* as a network with less temporal scales?

The motivation behind Transformer-QL is to increase the context length processed beyond what other methods can. I agree with the authors that extending the context is important. How effective is the method to capture farther long-term dependencies compared to previous methods? The experimental results mainly address similar networks with similar context lengths.
This could be tested by ablating elements from the input or ablating the recurrent memory vectors (setting them to zero during inference).

What would happen to the network if compression is removed?

==================

My score is based on the clarity improvements needed and the lack of evaluation of the effectiveness on the main objective of the method (longer contexts). I would appreciate if the authors can reply to my questions above.

==================

Minor issues:

-Section 2.1: “input sequence are” -> “input sequences are”

---

### Official Review · AnonReviewer4 · 2020-10-28
**Review #4**

**Rating:** 4
**Confidence:** 4

**Review:**

The authors propose Transformer-QL to capture the contextual information in multiple temporal scales - finer scales to capture recent past information and coarser scales to capture distance past information. The results show significant improvement in the perplexity score over Transformer-XL and Compressive Transformer. I agree that making use of multiple temporal scales can capturing more context information. However, the proposed method is like to merge the key ideas from Transformer-XL and Compressive Transformer. And my major concern regards the experiments. The performance of Transformer-QL is still far from the work "ADAPTIVE INPUT REPRESENTATIONS FOR NEURAL LANGUAGE MODELING" which does not target capturing long context. Then why do we need Transformer-QL? The authors should have done more experiments on other datasets, maybe refer to the paper, Big Bird: Transformers for Longer Sequences.

Pros:
1. The idea of using multiple temporal scales to capture more context information is convincing.
2. The experiments show that Transformer-QL can outperformance Transformer-XL and Compressive Transformer.

Cons:
1. The authors only test the model on one language modeling task and far from SOTA model on this task. More datasets should be used to evaluate the model.
2. The authors should have some comparison to some recent models on processing long sequences, such as the models from "Efficient Transformers: A Survey"

#######
As no author response, I will keep my rating.

---

### Official Review · AnonReviewer1 · 2020-10-28
**Solid incremental improvement to long-context Transformers**

**Rating:** 7
**Confidence:** 4

**Review:**

This paper proposes Transformer-QL, an improvement over Transformer-XL architecture which allows to use longer context at reduced cost. The paper is well-written and its good experimental results are strengthened by theoretical complexity estimation.

The text of the paper, though well-structured and concise, requires some polishing and misspellings correction (for example, I am doubtful that the proposed architecture is endowed with the ability to bite while section D of the appendix claims that "Multi-scale Transformer has been widely bitten by Transformer-QL").

---

### Decision · Program_Chairs · 2021-01-07
**Final Decision**

**Decision:**

Reject

**Comment:**

This paper introduces Transformer-QL, a new variant of transformer networks that can process long sequences more efficiently. This is an important research problem, which has been widely studied recently. Unfortunately, this paper does not compare to such previous works (eg. see "Efficient transformers: A survey"), the only considered baselines being Transformer-XL and Compressive transformer. Moreover, the reviewers found the experimental section to be lacking, as the results are weak compared to existing work, and important ablation studies are missing. The authors did not provide a rebuttal. For these reasons, I recommend to reject the paper.